# Investigation of Physico-Chemical Stability and Aerodynamic Properties of Novel “Nano-in-Micro” Structured Dry Powder Inhaler System

**DOI:** 10.3390/mi14071348

**Published:** 2023-06-30

**Authors:** Petra Party, Rita Ambrus

**Affiliations:** Faculty of Pharmacy, Institute of Pharmaceutical Technology and Regulatory Affairs, University of Szeged, Eötvös Street 6, 6720 Szeged, Hungary; party.petra@szte.hu

**Keywords:** nanotechnology, pulmonary delivery, dry powder inhaler, meloxicam, stability test

## Abstract

Pulmonary drug transport has numerous benefits. Large surface areas for absorption and limited drug degradation of the gastrointestinal system are provided through the respiratory tract. The administration is painless and easy for the patient. Due to their better stability when compared to liquid formulations, powders have gained popularity among pulmonary formulations. In the pharmaceutical sector, quality assurance and product stability have drawn a lot of attention. Due to this, it was decided to perform a long-term stability study on a previously developed, nanosized dry powder inhaler (DPI) formulation that contained meloxicam. Wet milling was implemented to reduce the particle size, and nano spray-drying was used to produce the extra-fine inhalable particles. The particle diameter was determined using dynamic light scattering and laser diffraction. Scanning electron microscopy was utilized to describe the morphology. X-ray powder diffraction and differential scanning calorimetry were applied to determine the crystallinity. In an artificial lung medium, the in vitro dissolution was studied. The Andersen Cascade Impactor was used to investigate the in vitro aerodynamic characteristics. The stability test results demonstrated that the DPI formulation maintained its essential qualities after 6 and 12 months of storage. Consequently, the product might be promising for further studies and development.

## 1. Introduction

One of the first methods of medication administration identified is respiratory drug delivery [1]. The beneficial characteristics of the lung enable the administration of larger drug concentrations to the airways for enhanced efficacy and to limit adverse effects. These advantages include avoiding first-pass metabolism and enzymatic inactivation [2]. The administration is non-invasive, which enhances patient compliance. One of the most widely used methods for treating local respiratory conditions, such as chronic obstructive pulmonary disease, asthma, pneumonia, and chronic pulmonary infections, is pulmonary drug delivery [3,4].

Nebulizers, metered dosage inhalers (MDIs), soft mist inhalers (SMIs), and dry powder inhalers (DPIs) are the most frequently utilized pulmonary medication delivery devices. MDIs have a larger carbon footprint than DPIs. DPIs are also cost-effective compared to MDIs [5]. Hydrofluorocarbon propellants are used in MDIs, which are greenhouse gases that persist in the atmosphere for years. Since DPIs are absent of these propellants and produce 20 g CO_2_ equivalent per dose compared to 500 g CO_2_ equivalent for MDIs, they have lower greenhouse gas emission potential [6]. The inhalation helps the active ingredient to enter the respiratory tract. DPIs are also portable tools that make it simple for the patient to administer the formulation. On the other hand, education is essential for the correct usage of the products. Due to their solid form, DPIs have outstanding stability and do not require cold chain storage [4,7].

There is an increasing variety of commercially available dry powder inhalers, and these vary significantly in terms of their design, technical features, and other specific features. Some inhalers include properties that make them likely to be effective for a variety of patients, which may offer a certain level of ease for healthcare providers. However, there are numerous factors to take into consideration when choosing the most effective inhaler for individuals with lung disorders. In addition to the type of drug contained in the inhaler, factors such as the degree of clinical evidence supporting its efficacy and safety, doctor and patient preferences, technical features of the various inhalers, and the delivery and deposition of the fine particle dose to the lungs may be crucial to assisting the physician in choosing the best device for each patient in order to optimize their treatment [8].

Carrier-based and carrier-free systems are the two main categories into which DPIs can be categorized. Applying conventional carrier-based DPIs, drug deposition in the respiratory area is insufficient. The active ingredients in these systems are attached to the surface of a carrier, which is typically lactose. Optimizing the aerosolization of the products is essential since the potential of powders is the appropriate dispersion and deposition in the airways. To improve the therapeutic effect, novel carrier-free DPIs have been developed. In that case, a complex powder is formulated by combining the active pharmaceutical ingredient (API) with appropriate excipients [9].

The upper respiratory tract, which includes the mouth, larynx, and pharynx, and the lower respiratory tract, which includes the trachea, bronchi, and lungs, are the two primary divisions of the respiratory system. When moving from the trachea to the distal airways, the diameters of the airways decrease as they approach the lower region and alveoli of the lung, and their number simultaneously rises [1]. Generally, the lower part is the target for orally inhaled medications. Unwanted particle deposition in the upper area might have regional negative effects, such as localized discomfort, coughing, dysphonia, and infections [10]. Controlling the theoretical aerodynamic diameter at 1–5 µm is required for the successful transport of particles via the pulmonary route to the desired area in the lungs [11]. In the lower parts, regional deposition is also critical for effective drug delivery [10]. Extra-fine particles (1–2 µm) are suitable for reaching the deeper areas because they deposit significantly more in the smaller peripheral lung structures than in the upper regions [12,13].

Recently, attention has been focused on the inhalable, poorly water-soluble drugs, such as antibiotics and anti-inflammatory and antifungal agents, which would acquire elevated local concentration for improved therapeutic effectiveness. However, a poorly water-soluble drug cannot be efficiently absorbed in the lung since it dissolves slowly in the limited volume of the lining fluid. The undissolved particles may be removed through alveolar macrophage uptake and mucociliary clearance, which leads to a compromised therapeutic effect. Additionally, leftover particles that remain for a long time on the surface of the lung epithelium may cause lung irritation and inflammation [14].

The development of nano-embedded microparticles for pulmonary application has drawn a growing amount of attention in recent years [15]. The systems enable the combination of nano- and microparticle benefits. Nanoparticles have advantages for getting through biological barriers [16]. The overall dosage required is reduced due to the enhanced drug transport in mucus and biofilms [17]. Problems with pulmonary drug administration may be resolved by using nanoparticle delivery methods [18]. The application of innovative and effective products that contain nanoparticles may enhance various therapies [19,20]. Therefore, DPI formulation with enhanced dissolution and improved absorption is urgently required for the pulmonary delivery of water-insoluble drugs. 

One of the most important factors for guaranteeing the safety and efficacy of pharmaceutical medicines is stability [21]. Drug nanoparticle stability issues, such as crystal formation, sedimentation, and agglomeration, may occur during production, storage, transportation, and application [22]. In general, liquid formulations are less stable than solid dosage forms. However, the potential of aggregation should be taken into consideration in the case of solid forms while using nanosized API. For efficient therapy, it is necessary to maintain the quality-influencing properties of the products. Proper attention should be given to drug nanocrystal stability difficulties during the development of pharmaceutical products [23].

In our previous studies, wet milling and nano spray drying were used to prepare a carrier-free DPI product consisting of nanosized meloxicam (MX) [24]. The “nano-in-micro” DPI can target the smaller airways with the extra-fine particles (<2 μm) and increase the water solubility of the drug. The alveolar section of the lung is where the nano-sized active ingredient may exert its anti-inflammatory effect; therefore, our goal is to deliver a high percentage of the extra-fine particles there. The combined preparation method can create particles under 2 µm with narrow size distribution. The previous investigations of the product revealed that the nanosized MX particles are partially amorphized, improving drug release. In addition, the product demonstrated significant drug deposition in the lung in vitro. The present study focused on the long-term stability of the developed DPI powder that contains nanosized MX. A significant challenge related to the development of formulations in DPIs is their stability. Manufacturing processes, pharmaceutical engineering techniques, and storage conditions can significantly impact the physical and aerosol stability of inhalable particles The physical stability of the DPIs is frequently overlooked in the literature even though they are critical to the quality and performance of the inhalation powders [4]. As a result of the extensive investigation, we could offer the MX a new, innovative therapeutic application in the management of severe lung inflammation. 

## 2. Materials and Methods

### 2.1. Materials

Meloxicam (MX) (Egis Pharmaceuticals PLC., Budapest, Hungary) was used as the active ingredient. MX is a non-steroidal anti-inflammatory medication (NSAID) that selectively inhibits cyclooxygenase-2 (COX-2). MX is commercially available only in oral, intravenous, and intralesional delivery routes. In human therapy, osteoarthritis, and arthritis are currently the principal indications of MX [25]. However, COX-2 inhibitors could be used to treat pulmonary inflammation, such as SARS-CoV-2 infection, which induces the COX-2 expression and may help to control the lung inflammation and damage seen in COVID-19 patients [26,27,28]. It could be advantageous to improve and maintain the condition of patients with cystic fibrosis (CF), chronic obstructive pulmonary disease (COPD), and non-small-cell lung cancer (NSCLC) [29,30,31,32]. The pulmonary suitable additives were poly-vinyl-alcohol 4–98 (PVA), (Aldrich Chemistry, Darmstadt, Germany) and L-leucine (LEU), (AppliChem GmbH, Darmstadt, Germany). The list of potential excipients is limited to materials that are biocompatible or endogenous to the lung and can effectively be eliminated [33].

### 2.2. Preparation Method

A two-step preparation method was used. Firstly, the nanosuspension was prepared by wet-milling technology. The final microsized powders were obtained by spray drying the diluted suspension.

#### 2.2.1. Wet Milling

Milling is a common scalable method used in the pharmaceutical industry for particle size reduction to improve the solubility and subsequently the bioavailability of poorly water-soluble APIs [34,35,36]. During wet milling, the drug is suspended in a liquid medium, such as surfactants or/and polymers, to stabilize the drug particles [37]. In this work, PVA was applied to maintain the stability and uniqueness of the MX particles. The MX particles are coated with PVA, which inhibits the drug particles from aggregating together during the size reduction process [38,39]. MX containing nanosuspension was prepared as follows: 2 g MX and 8 g of 2.5% (w/w%) PVA solution were added in a planetary ball mill (Retsch PM 100; Retsch Gmbh, Haan, Germany). The following conditions were present: 20 g of zirconium-dioxide beads (d = 0.3 mm), 500 rpm, 60 min. Due to the increased energy supply during wet milling, the development of local amorphous regions is conceivable and thus could alter the surface properties. This metastable state may change during handling and over storage time, and it could induce re-crystallization of these amorphous regions and particle size changes post-production [40].

#### 2.2.2. Spray Drying

Spray drying is a particle-engineering technique that is used to produce respirable powders for drug delivery to the deep lung [41], which is utilized in both laboratory and industrial environments. Compared to other popular drying methods, such as lyophilization, it attracted a lot of interest since it is less expensive, requires less time, and does not involve freezing, which is a high energy-consuming process [42]. One notable advantage of the nano spray dryer is its capacity to produce ultra-fine dry powder forms for a variety of materials, including heat-sensitive ones, with minimal damage. The advantages of a spray-dried powder include easy handling and storage as well as enhanced resistance to various environmental factors (such as light, oxidation, and temperature) [43]. LEU was incorporated into the MX nanosuspension to produce the dry powder. To enhance the aerosol characteristics, the administered dosage, and the overall efficacy of the product, LEU is frequently utilized as a dispersibility enhancer to minimize interparticle cohesion in DPIs. On the surface of spray-dried particles, LEU develops a crystalline layer, which reduces surface energy while enhancing surface rugosity [41]. The LEU enrichment at the surface of the particles can result in moisture protection, therefore improving their physical storage stability [44]. The inhalable powder was made by a Büchi Nano Spray Dryer (Büchi, Flawil, Switzerland) with a small nebulizer. Our preliminary research determined the following settings for the nano spray-drying process: inlet temperature of 80 °C, aspirator capacity of 100%, airflow rate of 120 mL/min, and pump rate of 20%. During the rapid drying of the method, the development of an amorphous form is possible and can lead to recrystallization. In the case of small-molecule drugs, such as MX, they tend to crystallize during spray drying [45].

#### 2.2.3. Physical Mixture Preparation

A physical mixture (PM) was created from the raw materials. The composition of the PM was similar to the spray-dried sample. During the experiments, the different qualities of the spray-dried samples were compared to the PM. Table 1 shows the compositions of the physical mixture and the final formulation.

### 2.3. Stability Test

Stability tests were performed at 25 ± 2 °C with 50 ± 5% relative humidity to imitate room conditions in a desiccator. Throughout the testing period, samples were stored in a closed glass jar. Samples were taken and measured after 1 day, 6 months, and 12 months [46,47].

### 2.4. Particle Size and Morphology Characterization

#### 2.4.1. Laser Diffraction

The particle sizes, the particle size distributions, and the specific surface areas of the samples were determined using laser diffraction (Malvern Mastersizer Scirocco 2000, Malvern Instruments Ltd., Worcestershire, UK). The refractive index of MX was adjusted to 1.720. MX was given a new refractive index of 1.720. The dry dispersion equipment was used to examine the nano spray-dried particles. The 3.0 bar of dispersion air pressure and 75% vibration feed were employed. Each sample was measured in triplicate. The particle size distribution (PSD) was described using the values D [0.1] (10% of the volume distribution is below this value), D [0.5] (50% of the volume distribution is below this value), and D [0.9] (90% of the volume distribution is below this value) (Equation (1)). Span values were indicated in the particle size distribution, the larger the span value, the broader the spread. The specific surface area (SSA) was calculated using the PSD data. The computations were performed assuming the particles were spherical.
(1)$$\mathrm{Span}=\frac{D\left[0.9\right]-D\left[0.1\right]}{D\left[0.5\right]},$$

#### 2.4.2. Dynamic Light Scattering (DLS)

Malvern Zetasizer Nano ZS (Malvern Instruments, Worcestershire, UK) was applied to analyze the average hydrodynamic diameter (d), polydispersity index (PDI), and zeta potential (ζ pot.) using DLS. The powders were dispersed in purified water and measured in folded capillary cells at 25 °C. A refractive index of 1.720 was assigned to MX. Each investigation was performed three times.

#### 2.4.3. Scanning Electron Microscopy (SEM)

The shape and size of the particles were analyzed using SEM (Hitachi S4700; Hitachi Ltd., Tokyo, Japan) operating at a 10 kV voltage. Using a sputter coater (Bio-Rad SC502; VG Microtech, Uckfield, UK) and a 2.0 kV at 10 mA electric potential for 10 min, the samples received a 90-s gold-palladium coating. The range of air pressure was 1.3 to 13.0 mPa. ImageJ, a free and open-source image analyzer was used to perform the particle diameter analysis. (https://imagej.nih.gov/ij/index.html, accessed on 22 June 2023.).

### 2.5. Structural Investigation

#### 2.5.1. Differential Scanning Calorimetry (DSC)

Thermal analyses of samples were performed using a Mettler Toledo TGA/DSC thermal analysis equipment (Mettler-Toledo GmbH, Greifensee, Switzerland). The DSC measurements were performed by analyzing 3–5 mg of samples heated to temperatures between 25 and 300 °C at a rate of 10 °C/min while maintaining a steady flow of argon at a rate of 10 L/h. The STAR^e^ program was used to analyze the data (Mettler-Toledo GmbH, Greifensee, Switzerland). The ratio normalized integrals were used to calculate the crystallinity indexes with the PM samples being considered as 100%.

#### 2.5.2. X-ray Powder Diffraction (XRPD)

The crystal structure and level of crystallinity were measured using XRPD. Based on the X-ray diffraction concept, a sharp peak is seen when X-rays diffract from atoms that repeatedly appear in the same position within the unit cell (i.e., a crystalline structure). The BRUKER D8 advance X-ray powder diffractometer (Bruker AXS GmbH, Karlsruhe, Germany) and VNTEC-1 detector (Bruker AXS GmbH, Karlsruhe, Germany) were used to perform the XRPD measurement. The powder samples were placed on a slide of flat quartz glass with an etched square and measured. At 40 kV and 40 mA, samples were scanned. With a step time of 0.1 s and a step size of 0.007°, the angular range was 3–40°. The DIFFRACplus EVA program was used for all manipulations, including K2 stripping, background removal, and smoothing of the area under the diffractogram peaks. Based on Equation (2), the values for the crystallinity index (X_c_) were determined. The PM sample was considered 100% crystalline. A symbolizes the area under the curve:(2)$$X_{c}=\frac{A_{crystalline}}{A_{crystalline}+A_{amorphous}}\times 100\;$$

### 2.6. In Vitro Drug Release Study

Currently, there are no standardized methodologies or regulatory criteria to evaluate in vitro dissolution tests of inhaled powders [48]. Using a modified paddle method, the rate of drug release of powders was examined (Hanson SR8 Plus, Teledyne Hanson Research, Chatsworth, CA, USA). The analysis was performed under pulmonary circumstances (37 °C and pH: 7.4). The following components were included in 50 mL of artificial lung medium: 2.27 g/L NaHCO_3_, 0.68 g/L NaCl, 0.1391 g/L NaH_2_PO_4_, 0.02 g/L CaCl_2_, and 5.56 mL/L 0.1 M H_2_SO_4_ and 0.37 g/L glycine [49]. The sample preparation was completed after 5, 10, 15, 30, 45, and 60 min. The paddle rotated at 100 rotations per minute. To maintain the permanent volume constant, lung fluid was simultaneously added at every point of sampling to replace 5 mL of the sample. For filtering, cellulose ester membranes with 0.22 m pore sizes were employed. Following filtration, spectrophotometry at 362 nm (Unicam UV/VIS Spectrophotometer, Cambridge, UK) was used to determine the drug content of the aliquots. The experiments were performed in three sets.

### 2.7. In Vitro Aerodynamic Performance

Using an Andersen Cascade Impactor (ACI, Apparatus D, Copley Scientific Ltd., Nottingham, UK), (Figure 1), which is authorized by the European Pharmacopoeia [50], it was determined how effectively the formulations in vitro aerosolized. The inhalation flow rate was set to 60 l/min (high-capacity pump model HCP5, critical flow controller model TPK, Copley Scientific Ltd., Nottingham, UK). To calculate the actual flow rate, a mass flow meter (flow meter model DFM 2000, Copley Scientific Ltd., Nottingham, UK) was utilized. There was a 4-s breathe-in. The setup imitates a regular, healthy breathing rhythm with a 4 L inhalation volume. The powders were delivered using Breezhaler^®^ single-dose devices (Novartis International AG, Basel, Switzerland) and size 3 gelatin capsules (Capsugel, Bornem, Belgium). It is a breath-activated device that administers drugs for asthma and chronic obstructive pulmonary disease. To simulate the conditions of pulmonary adhesion, the collecting plates on the stages were coated with a mixture of Span 85 and cyclohexane (1 + 99 w/w%). Following inhalation, the apparatus, capsules, induction port, plates, and filter were rinsed with methanol and pH 7.4 phosphate buffer (60 + 40 V/V%) to collect and dissolve the deposited MX. The API was measured using an ATI-UNICAM UV/VIS Spectrophotometer (Cambridge, UK) at a wavelength of 362 nm. The in vitro aerodynamic properties were evaluated with the support of the data analysis tool, Inhalytix^TM^ (Copley Scientific Ltd., Nottingham, UK)*,* which is a completely approved and verified aero-dynamic particle size distribution data analysis solution. The most frequently employed parameter is fine particle fraction (FPF). FPF is calculated as the percentage of the mass of the active ingredient containing particles with an aerodynamic diameter of less than 5 μm divided by the emitted dose of the formulations. Additionally, the emitted fraction (EF), also known as the released fraction from the DPI device, was calculated.

### 2.8. Statistical Analysis

The statistical analysis was carried out using GraphPad Prism 8.0.1 (GraphPad Software, CA, USA). The Student’s *t*-test was employed to assess the statistical significance. Changes were determined to be statistically significant at *p* = 0.05.

## 3. Results and Discussion

### 3.1. Characterization of Particle Size and Morphology

#### 3.1.1. Laser Diffraction

The original particle of the MX was D [0.5] = 9.913 ± 0.371 μm. As a result of wet milling, the particle size of API in the suspension decreased to D [0.5] = 137.70 nm ± 4.965 nm. The reduced particle size and improved surface area of the particles lead to a faster dissolution rate [51]. After nano spray-drying, the D [0.5] values of the samples were 1.429 ± 0.09 μm. For the treatment of deeper lung segments, the use of extra-fine built-up nanosized API could be advantageous [18,52]. According to the further particle size analysis, the particles of the products remained the same particle size during the testing period of the study (Table 2), proving that the particles did not agglomerate. The size was within the required 1 to 5 µm particle size range for pulmonary delivery [53]. The statistical analysis revealed that there was no noticeable variation in the particle size formulations over the storage period. The span values were between 1.3 and 1.6, which means that the distribution of the particles was monodisperse [54]. The monodisperse particle size distribution is crucial for proper lung deposition and dosage uniformity. The specific surface area (SSA) numbers slightly increased, suggesting that the dissolution of the drug may be improved.

#### 3.1.2. Dynamic Light Scattering

Table 2 shows the outcomes of the DLS investigation. DLS measured the liquid suspension of the particles. The particles were deagglomerated and resulted in decreased particle sizes compared to the laser diffraction. The diameters of the dispersed nano spray-dried particles were between 500 and 700 nm. It predicts that as the dry particles deposit into the lung fluid and disintegrate into nanoparticles, resulting in larger dissolution and cellular absorption [52]. Although there was an increase in particle diameter, the dry particle size is far more important in terms of lung deposition. Inhomogeneous distribution is shown by the relatively high PDI results (PDI > 0.3). However, it is not considered a problem if it does not negatively affect the drug release [39]. The ζ potential values were slightly decreased with time (Table 3). The systems are more degradable and less retentive in the airways due to the negative ζ potential values of the spray-dried formulations. It is beneficial since the particles are unlikely to trigger further fibrosis, infection, or inflammation [55,56].

#### 3.1.3. Scanning Electron Microscopy

The stored samples were studied by SEM to understand the morphology of the particles. The SEM images of the spray-dried formulations showed spherical morphology (Figure 2). The optimized nano spray-drying created the spherical particle form [57]. The spray-dried spherical particles have a smaller contact area and a more uniform particle size distribution, which leads to a larger FPF compared to mechanically micronized drugs [58]. On the PVA surface, the MX nanoparticles were seen in a homogeneous distribution. The MX particles were covered with PVA, which helped their separation from one another [59]. Throughout the testing period, they showed no signs of aggregation. A slightly rougher surface can be observed because the humidity caused moisture absorption [60]. The results of the laser diffraction method were similar to the results of the ImageJ 1.53e software, which was utilized to determine the particle size based on the pictures (Table 4). The particle size of around 1 μm is beneficial for deep pulmonary delivery.

### 3.2. Structural Investigation

#### 3.2.1. Differential Scanning Calorimetry (DSC)

The DSC curves (Figure 3) throughout the testing period demonstrate that the partially amorphous characteristic of the samples did not change significantly. The original melting point of the MX was 246.55 °C; however, milling had the effect of lowering it to 240.87 °C. In the case of the DPI formulation, the character and size of the endothermic peak are nearly identical at each sample point (6 months: 238.46 °C, 12 months: 236.69 °C). The degrees of crystallinity of the samples ranged from 53 to 70% when compared to the PM, which was taken as 100% (Table 5).

#### 3.2.2. X-ray Powder Diffraction (XRPD)

If the XRPD patterns of MX and the chosen excipients are known, XRPD can be used to observe the structural changes of the DPI samples throughout storage. Particularly, the properties of the active ingredient could be crucial because their crystalline or amorphous forms may result in morphological differences and affect the interactions between the particles, which could have an impact on the aerodynamic results. MX has characteristic peaks with the highest intensities at 6.6°, 11.4°, 13.1°, 13.5°, 15.1°, 18.7°, 19.3°, 25.9°, and 26.4° 2-theta peaks, indicating its crystalline structure. We detected the characteristic peaks of LEU at 6.12°, 24.39°, and 30.61° 2-theta peaks. The presence of PVA did not affect the diffractograms. The intensity of the peaks decreased in the 1-day sample as a result of the milling, and the crystallinity of the MX also decreased. This decreased crystallinity persisted at 6 and 12 months (Figure 4). The 1-day sample had a 49.81% crystallinity index, which did not change significantly throughout storage. The crystallinity indexes of the samples are shown in Table 5. The results of the measurement supported the DSC findings, indicating that the milling impact and the presence of PVA throughout the testing period caused the MX to become and remain partially amorphous.

### 3.3. In Vitro Drug Release Study

Figure 5 shows that the MX released quickly and in larger quantities in the case of the DPI samples than the PM. It is beneficial because the dissolution rate is the rate-limiting step for absorption of MX. The nanosized form, the increased surface area, and the partial amorphization of MX promoted a faster dissolution rate [61]. Furthermore, more API was released from the spray-dried samples than the PM because PVA prevented particle aggregation and LEU reduced cohesion between the particles. The rapid release was maintained. Furthermore, the drug release from stored samples was faster, possibly the water absorption on the surface helped the wettability of the particles. The outcomes are beneficial in local treatment because the mucociliary clearance has minimal time to eliminate the particles [14,62].

### 3.4. In Vitro Aerodynamic Properties

The distribution of the spray-dried samples was established during the aerodynamic analysis. Figure 6 illustrated how the samples were deposited on different areas of the set. An efficient MX deposition was on the third and fourth stages. The majority of the formulations reached the filter, which models the deeper lung region [63]. Successfully, the powder targeted the smaller airways even at the end of the stability study. The calculated in vitro aerodynamic results by Inhalytix™ V 2.0 software are presented in Table 6. The samples showed outstanding FPF values (93–95%), which exceed the FPF values of the commercially available formulations in the Breezhaler^®^ inhalator. The high emitted fraction (EF) showed an insignificant decrease. The application of LEU improved the aerosolization of the products owing to the increased cohesion between the particles of the powder and the adhesion between the powder and the gelatine capsule.

## 4. Conclusions

In this study, the stability test of a carrier-free, novel DPI sample containing nanosized, non-steroidal, anti-inflammatory drug was examined at normal room conditions. After the storage, the formulation presented advantageous characteristics, thanks to the technological steps and the compositions. Wet media milling is one of the most popular methods in the pharmaceutical field to produce stable nanosuspension of poorly water-soluble APIs. To optimize time and cost considerations and accurately predict milling performance at higher scales, a variety of modeling techniques could be used in the industry. The spray-drying technique has also been successfully applied at both laboratory and industrial scales. The advantages of a spray-dried powder include easy use and long shelf-life stability. The development of novel delivery methods can be a strategy for repositioning medications; therefore, it saves money and time for the pharmaceutical industry. The pulmonary route of MX could be an intriguing solution for treating different lung inflammation, which can be caused by SARS-CoV-2 infection, CF and COPD, and NSCLC. The excipients were pulmonary-approved materials. During the testing period, the particle size remained unchanged, while the particle size of MX in the formulation increased but did not differ significantly. Furthermore, the partially amorphous property of MX persisted throughout the stability examination. The outcomes of the dissolution test demonstrated that the initially large amount of drug was released from the samples in the examination time. The results suggest that PVA might inhibit the particles from aggregation and crystallizing. The aerosol performance of the formulated DPIs did not deteriorate. The sample has beneficial FPF and EF results after 12 months. The addition of LEU enhanced the aerosolization of the products. The outcome of this study demonstrates that the “nano-in.micro” DPI can maintain its quality for an extended period. According to ICH guidelines, further stability investigations are required, such as a test of the final package.

## Figures and Tables

**Figure 1 micromachines-14-01348-f001:**
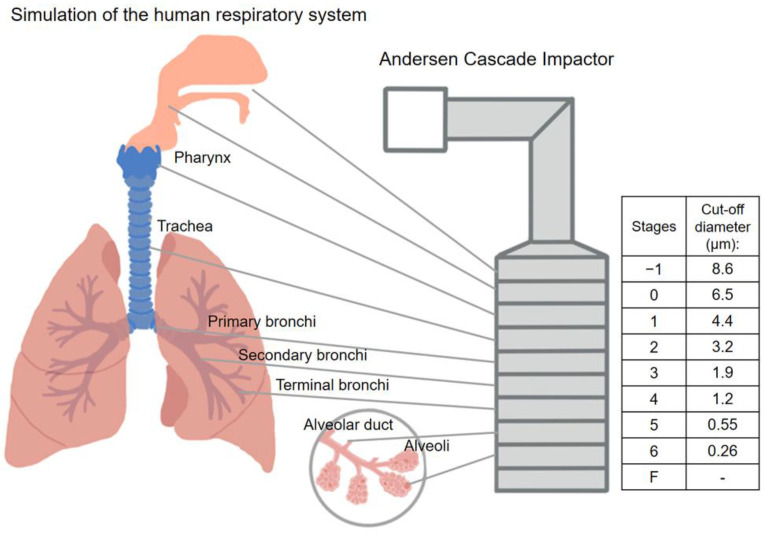
Correlation between the human respiratory system and the Andersen Cascade Impactor.

**Figure 2 micromachines-14-01348-f002:**
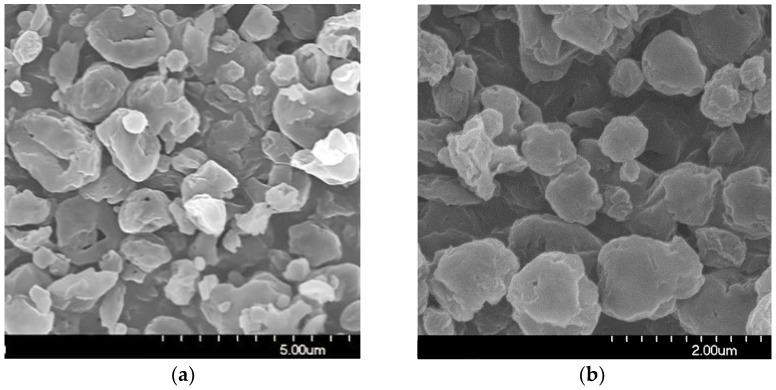
SEM records of the samples. (**a**) 1 day; (**b**) 1 day, (**c**) 6 months; (**d**) 6 months, (**e**) 12 months; (**f**) 12 months.

**Figure 3 micromachines-14-01348-f003:**
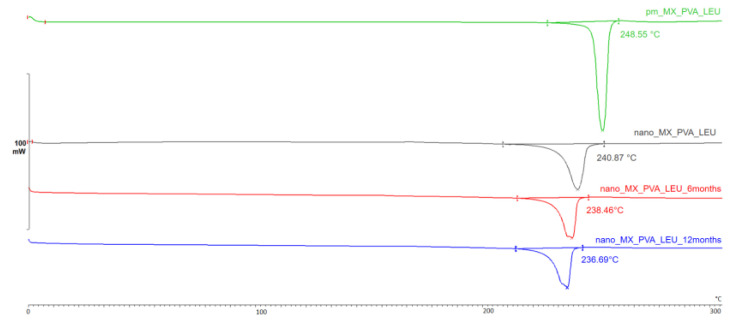
DSC curves of the samples.

**Figure 4 micromachines-14-01348-f004:**
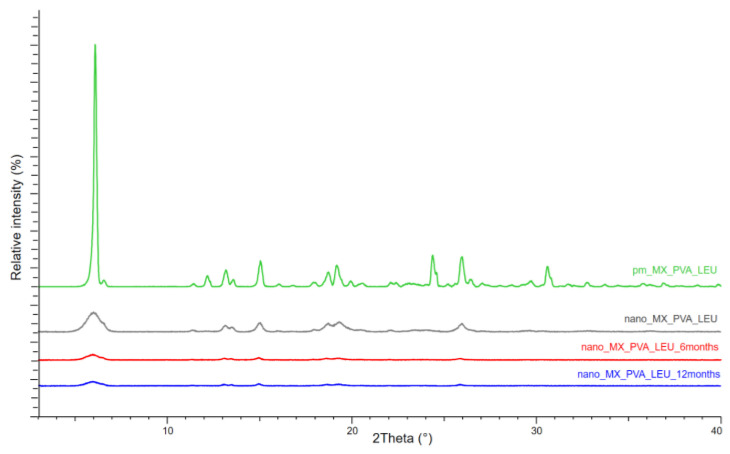
XRPD results of the samples.

**Figure 5 micromachines-14-01348-f005:**
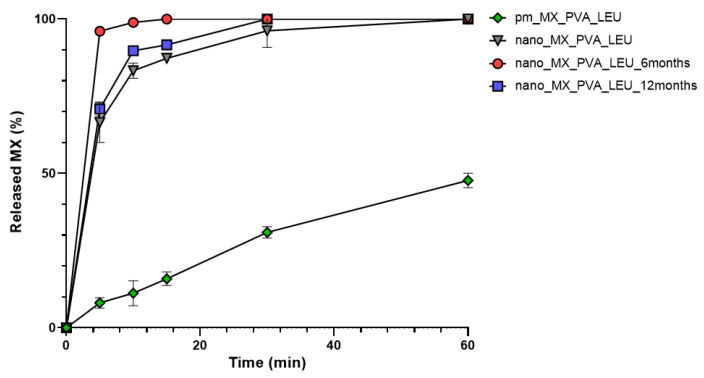
In vitro dissolution of the spray-dried samples and the physical mixture.

**Figure 6 micromachines-14-01348-f006:**
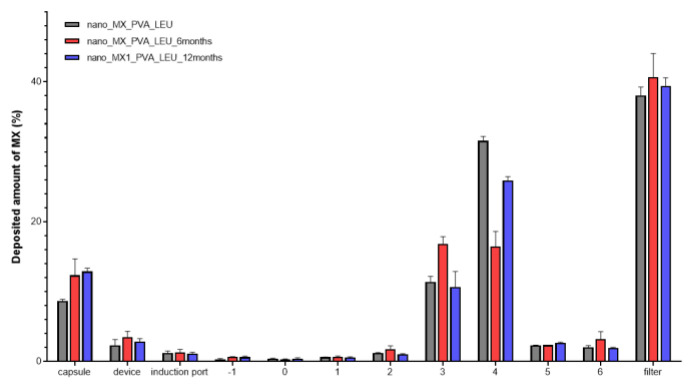
Distribution of the deposited MX in the ACI.

**Table 1 micromachines-14-01348-t001:** The composition of the physical mixture and the spray-dried sample.

Sample	MX (g)	PVA (g)	LEU (g)
pm_MX_PVA_LEU	2.00	0.45	2.00
nano_MX_PVA_LEU	2.00	0.45	2.00

**Table 2 micromachines-14-01348-t002:** Particle sizes, spans, and SSA results of the samples.

Samples	D [0.1] (μm)	D [0.5] (μm)	D [0.9] (μm)	Span	SSA (m^2^/g)
1 day	0.789 ± 0.02	1.429 ± 0.09	3.400 ± 0.50	1.586 ± 0.12	4.385 ± 0.01
6 months	0.800 ± 0.01	1.471 ± 0.05	3.110 ± 0.77	1.562 ± 0.47	4.445 ± 0.26
12 months	0.749 ± 0.01	1.441 ± 0.02	2.703 ± 0.03	1.357 ± 0.01	4.725 ± 0.06

Data are means ± SD (*n* = 3 independent measurements).

**Table 3 micromachines-14-01348-t003:** Average particle diameter, polydispersity index, and **ζ** potential of samples.

Samples	d (nm)	PDI	ζ pot. (mV)
1 day	526.90 ± 20.0	0.381 ± 0.03	−24.50 ± 1.47
6 months	548.57 ± 35.9	0.382 ± 0.03	−23.30 ± 7.91
12 months	657.37 ± 46.9	0.496 ± 0.07	−22.23 ± 0.55

Data are means ± SD (*n* = 3 independent measurements).

**Table 4 micromachines-14-01348-t004:** The particle sizes of the spray-dried samples according to the SEM images.

Samples	d (μm)
1 day	1.106 ± 0.29
6 months	1.240 ± 0.40
12 months	1.060 ± 0.19

Data are means ± SD (*n* = 100 independent measurements).

**Table 5 micromachines-14-01348-t005:** The crystallinity index (X_c_) of the samples.

Samples	X_c_ (%)	
	DSC	XRPD
PM	100	100
1 day	69.92	49.81
6 months	53.91	42.51
12 months	53.67	41.89

**Table 6 micromachines-14-01348-t006:** FPF and EF of the spray-dried samples.

Samples	FPF by Size (%)	FPF by Stage (%)	EF (%)
1 day	94.94 ± 0.39	95.35 ± 0.39	89.04 ± 1.08
6 months	92.90 ± 0.80	93.39 ± 0.71	84.17 ± 3.20
12 months	93.85 ± 0.31	94.28 ± 0.38	84.27 ± 0.88

Data are means ± SD (*n* = 3 independent measurements).

## Data Availability

We declare the real scientific content of our data.

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
