# Peer review of "Investigation of Physico-Chemical Stability and Aerodynamic Properties of Novel “Nano-in-Micro” Structured Dry Powder Inhaler System"

_micromachines, 2023, doi:10.3390/mi14071348_

Round 1

Reviewer 1 Report

Petra Party et al. reported an interesting work upon a nano-in-micro DPI. Although this work fell within the scope of our journal, it could not satisfy the standard in the current form. The reviewer suggested to reconsider this work after a resubmission. There were several issues pending addressed:

1)      The novelty of the DPI system needed to be demonstrated. The used excipients (PVA, LEU, etc.) were commonly used materials for DPI, and the concept “nano-in-micro” was also a comprehensively investigated one. As a result, what’s new in this work?

2)      ACI was employed for aerodynamic tests in this work. Currently, the gold-standard has become NGI, instead of ACI. Thus, it would be better to utilize NGI for aerodynamic tests.

3)      According to Table 3, the PDI values of nanoparticles were quite high (> 0.3), which suggested that the size distribution was not homogeneous. The authors should re-perform the tests or make proper discussions.

4)      The significant digits of Table 2, 3, 4, 5 should be unified in each table.

5)      A short Discussion Section was presented, pending expansion. For example, the clinical translation and industrialization aspects could be discussed.

6)      A Conclusion Section was missing. Please supplement.

7)      Please check the format of Ref #17 and #26.

Author Response

Reviewer 1

Thank you very much for your opinion. Below are listed all of the modifications made in the paper, according to the suggestions, and you can find them with orange color in the text.

Comments and Suggestions for Authors

Petra Party et al. reported an interesting work upon a nano-in-micro DPI. Although this work fell within the scope of our journal, it could not satisfy the standard in the current form. The reviewer suggested to reconsider this work after a resubmission. There were several issues pending addressed.

  1. The novelty of the DPI system needed to be demonstrated. The used excipients (PVA, LEU, etc.) were commonly used materials for DPI, and the concept “nano-in-micro” was also a comprehensively investigated one. As a result, what’s new in this work?

Thank you for the comment. The introduction was extended. About the stability of the “nano-in-micro” DPIs there are insufficient amount of available information, however it is one of the most crucial factors in case of nanoparticles. The new information in this work is the validation of the long term-stability of these systems, not the application of new excipients

In our previous studies, wet milling and nano spray drying were used to prepare a carrier-free DPI product consisting of nanosized meloxicam (MX) [24]. The “nano-in-micro” DPI can target the smaller airways with the extra-fine particles (<2 μm) particles and increased the water-solubility of the drug. The alveolar section of the lung is where the nano-sized active ingredient may exert its anti-inflammatory effect, therefore our goal is to deliver a high percentage of the extra-fine particles there. The combined preparation method can create particles under 2 µm with narrow size distribution. The previous investigations of the product revealed that the nanosized MX particles are partially amorphized, improving drug release. In addition, the product demonstrated significant drug deposition in the lung in vitro. The present study focused on the long-term stability of the developed DPI powder that contains nanosized MX. A significant challenge related to the development of formulations in DPIs is their stability. Manufacturing processes, pharmaceutical engineering techniques, and storage conditions can significantly impact the physical and aerosol stability of inhalable particles The physical stability of the DPIs is frequently overlooked in the literature even though they are critical to the quality and performance of the inhalation powders [4]. As a result of the extensive investigation, we could offer the MX a new, innovative therapeutic application in the management of severe lung inflammation.

  1. ACI was employed for aerodynamic tests in this work. Currently, the gold-standard has become NGI, instead of ACI. Thus, it would be better to utilize NGI for aerodynamic tests.

Thank you for the comment. Yes, NGI is the standard method, but our research team has widespread experience using the ACI. ACI is authorized by the European Pharmacopoeia [2] and United States Pharmacopeia [3] for aerodynamic assessment. For further investigation to imitate the insufficient breathing pattern of the patient, ACI could be a better option for the measurements using lower flow-rate [4], [5].

Using an Andersen Cascade Impactor (ACI, Apparatus D, Copley Scientific Ltd., Nottingham, United Kingdom), (Figure 1), which is authorized by the European Pharma-copoeia [50], it was determined how effectively the formulations in vitro aerosolized.

  1. According to Table 3, the PDI values of nanoparticles were quite high (> 0.3), which suggested that the size distribution was not homogeneous. The authors should re-perform the tests or make proper discussions.

Thank you for the comment. The discussion of the PDI result was modified.

As DLS measured the liquid suspension of the particles. The particles were deagglom-erated and resulted in decreased particle sizes compared to the laser diffraction. The diameters of the dispersed nano spray-dried particles were between 500-700 nm. It predicts that as the dry particles deposit into the lung fluid, and disintegrate into na-noparticles, resulting in larger dissolution and cellular absorption [52]. Although there was an increase in particle diameter, the dry particle size is far more important in terms of lung deposition. Inhomogeneous distribution is shown by the relatively high PDI results (PDI>0.3). However, it is not considered a problem if it does not negatively affect the drug release [39].

  1. The significant digits of Table 2, 3, 4, 5 should be unified in each table.

Thank you for the comment, the tables are modified.

  1. A short Discussion Section was presented, pending expansion. For example, the clinical translation and industrialization aspects could be discussed.

Thank you for the comment. The discussion was extended merged with the results. The conclusion was extended with the clinical translation and the industrialization. They were also mentioned in the materials and the preparation method sections.

Meloxicam (MX) (Egis Pharmaceuticals PLC., Budapest, Hungary) was used as the active ingredient. MX is a non-steroidal anti-inflammatory medication (NSAID) that selectively inhibits cyclooxygenase-2 (COX-2). MX is commercially available only in oral, intravenous, and intralesional delivery routes. In human therapy, osteoarthritis, and arthritis are currently the principal indications of MX [25]. However, COX-2 inhibitors could be used to treat pulmonary inflammation, such as SARS-CoV-2 infection, which induces the COX-2 expression and may help to control the lung inflammation and damage seen in COVID-19 patients [26–28]. It could be advantageous to improve and maintain the condition of patients with cystic fibrosis (CF), chronic obstructive pulmonary disease (COPD), and non-small-cell lung cancer (NSCLC) [29–32]. The pulmonary suitable additives were poly-vinyl-alcohol 4–98 (PVA), (Aldrich Chemistry, Darmstadt, Germany) and L-leucine (LEU), (AppliChem GmbH, Darmstadt, Germany). The list of potential excipients is limited to materials that are biocompatible or endogenous to the lung and can effectively be eliminated [33].

Milling is a common scalable method used in the pharmaceutical industry for particle size reduction to improve the solubility and subsequently bioavailability of poorly water-soluble APIs [34–36]. During wet milling, the drug is suspended in a liquid medium, such as surfactants or/and polymers to stabilize the drug particles [37].

Spray-drying is a particle engineering technique that is used to produce respirable powders for drug delivery to the deep lung [41], which is utilized in both laboratory and industrial environments. Compared to other popular drying methods like lyophilization, it attracted a lot of interest since it is less expensive, requires less time, and does not involve freezing, which is a high energy-consuming process[42]. One notable advantage of the nano spray-dryer is its capacity to produce ultra-fine dry powder forms for a variety of materials, including heat-sensitive ones, with minimal damage. The advantages of a spray-dried powder include easy handling and storage as well as enhanced resistance to various environmental factors (such as light, oxidation, and temperature)[43].

  1. A Conclusion Section was missing. Please supplement.

Thank you for the comment. The conclusion section was added.

In this study, the stability test of a carrier-free, novel DPI sample containing nanosized non-steroidal anti-inflammatory drug was examined at normal room conditions. After the storage, the formulation presented advantageous characteristics, thanks to the technological steps and the compositions. Wet media milling is one of the most popular method in the pharmaceutical field to produce stable nanosuspension of poorly water-soluble APIs. To optimize time and cost considerations and accurately predict milling performance at higher scales, a variety of modeling techniques could be used in the industry. The spray-drying technique has also been successfully applied at both laboratory and industrial scales. The advantages of a spray-dried powder include easy use and long shelf life stability. The development of novel delivery methods can be a strategy for repositioning medications, therefore it saves money and time for the pharmaceutical industry. The pulmonary route of MX could be an intriguing solution for treating different lung inflammation, which can be caused by SARS-CoV-2 infection, CF and COPD, and NSCLC. The excipients were pulmonary approved materials. During the testing period, the particle size of the remained unchanged, while the particle size of MX in the formulation increased, but did not differ significantly. Furthermore, the partially amorphous property of MX persisted throughout the stability examination. The outcomes of the dissolution test demonstrated, that the initially large amount of drug was released from the samples in the examination time. The results suggest that PVA might inhibit the particles from aggregation and crystallizing. The aerosol performance of the formulated DPIs did not deteriorate. The sample has beneficial FPF and EF results after 12 months. The addition of LEU enhanced the aerosolization of the products. The outcome of this study demonstrates, that the “nano-in.micro” DPI can maintain its quality for an extended period. According to ICH guidelines, further stability investigations are required, such as a test in the final package.

  1. Please check the format of Ref #17 and #26.

Thank you for the comment. The citations are modified.

  1. Shetty, N.; Cipolla, D.; Park, H.; Zhou, Q.T. Physical stability of dry powder inhaler formulations. Expert Opin. Drug Deliv. 2020, 17, 77–96, doi:10.1080/17425247.2020.1702643.
  2. 2.9.18. Preparations for inhalation: aerodynamic assessment of fine particles. In European Pharmacopoeia 10.0; Strasbourg, 2019; p. 354 ISBN 9789287189127.
  3. Physical tests and determinations: aerosols. In United States Pharmacopeia; Rockville, 2010.
  4. Mohammed, H.; Arp, J.; Chambers, F.; Copley, M.; Glaab, V.; Hammond, M.; Solomon, D.; Bradford, K.; Russell, T.; Sizer, Y.; et al. Investigation of Dry Powder Inhaler (DPI) resistance and aerosol dispersion timing on emitted aerosol aerodynamic particle sizing by multistage cascade impactor when sampled volume is reduced from compendial value of 4 L. AAPS PharmSciTech 2014, 15, 1126–1137, doi:10.1208/s12249-014-0111-1.
  5. Taki, M.; Marriott, C.; Zeng, X.M.; Martin, G.P. Aerodynamic deposition of combination dry powder inhaler formulations in vitro: A comparison of three impactors. Int. J. Pharm. 2010, 388, 40–51, doi:10.1016/j.ijpharm.2009.12.031.
  6. Andrade, F.; Rafael, D.; Videira, M.; Ferreira, D.; Sosnik, A.; Sarmento, B. Nanotechnology and pulmonary delivery to overcome resistance in infectious diseases. Adv. Drug Deliv. Rev. 2013, 65, 1816–1827, doi:10.1016/j.addr.2013.07.020.
  7. Meloxicam - Drugbank Available online: https://go.drugbank.com/drugs/DB00814.
  8. Chen, J.S.; Alfajaro, M.M.; Wei, J.; Chow, R.D.; Filler, R.B.; Eisenbarth, S.C.; Wilen, C.B. Cyclooxgenase-2 is induced by SARS-CoV-2 infection but does not affect viral entry or replication. bioRxiv 2020, doi:10.1101/2020.09.24.312769.
  9. Ong, S.W.X.; Tan, W.Y.T.; Chan, Y.H.; Fong, S.W.; Renia, L.; Ng, L.F.P.; Leo, Y.S.; Lye, D.C.; Young, B.E. Safety and potential efficacy of cyclooxygenase-2 inhibitors in coronavirus disease 2019. Clin. Transl. Immunol. 2020, 9, 1–9, doi:10.1002/cti2.1159.
  10. Sarcinelli, M.A.; Martins da Silva, T.; Artico Silva, A.D.; Ferreira de Carvalho Patricio, B.; Mendes de Paiva, F.C.; Santos de Lima, R.; Leal da Silva, M.; Antunes Rocha, H.V. The pulmonary route as a way to drug repositioning in COVID-19 therapy. J. Drug Deliv. Sci. Technol. 2021, 63, doi:10.1016/j.jddst.2021.102430.
  11. Szabó-Révész, P. Modifying the physicochemical properties of NSAIDs for nasal and pulmonary administration. Drug Discov. Today Technol. 2018, 27, 87–93, doi:10.1016/j.ddtec.2018.03.002.
  12. Arafa, H.M.M.; Abdel-Wahab, M.H.; El-Shafeey, M.F.; Badary, O.A.; Hamada, F.M.A. Anti-fibrotic effect of meloxicam in a murine lung fibrosis model. Eur. J. Pharmacol. 2007, 564, 181–189, doi:10.1016/j.ejphar.2007.02.065.
  13. Yokouchi, H.; Kanazawa, K.; Ishida, T.; Oizumi, S.; Shinagawa, N.; Sukoh, N.; Harada, M.; Ogura, S.; Munakata, M.; Dosaka-Akita, H.; et al. Cyclooxygenase-2 inhibitors for non-small-cell lung cancer: A phase II trial and literature review. Mol. Clin. Oncol. 2014, 2, 744–750, doi:10.3892/mco.2014.319.
  14. Weiss, A.; Porter, S.; Rozenberg, D.; O’Connor, E.; Lee, T.; Balter, M.; Wentlandt, K. Chronic Obstructive Pulmonary Disease: A Palliative Medicine Review of the Disease, Its Therapies, and Drug Interactions. J. Pain Symptom Manage. 2020, 60, 135–150, doi:10.1016/j.jpainsymman.2020.01.009.
  15. Bilgili, E.; Guner, G. Mechanistic Modeling of Wet Stirred Media Milling for Production of Drug Nanosuspensions. AAPS PharmSciTech 2021, 22, doi:10.1208/s12249-020-01876-w.
  16. Seibert, K.D.; Collins, P.C.; Luciani, C. V.; Fisher, E.S. Milling operations in the pharmaceutical industry. Chem. Eng. Pharm. Ind. 2019, 861–879, doi:10.1002/9781119600800.ch38.
  17. Singare, D.S.; Marella, S.; Gowthamrajan, K.; Kulkarni, G.T.; Vooturi, R.; Rao, P.S. Optimization of formulation and process variable of nanosuspension: An industrial perspective. Int. J. Pharm. 2010, 402, 213–220, doi:10.1016/j.ijpharm.2010.09.041.
  18. Strojewski, D.; Krupa, A. Spray drying and nano spray drying as manufacturing methods of drug-loaded polymeric particles. Polim. Med. 2022, 52, 101–111, doi:10.17219/pim/152230.
  19. Salama, A.H. Spray drying as an advantageous strategy for enhancing pharmaceuticals bioavailability. Drug Deliv. Transl. Res. 2020, 10, 1–12, doi:10.1007/s13346-019-00648-9.

Reviewer 2 Report

The study represents an extension of the authors' previously published work on DPI formulation with MX, aiming to investigate the  physico-chemical stability and certain aerodynamic parameters of the formulation over a period of 3-6 months. Although the study is interesting, the authors' lack of meticulousness in completing the manuscript compared to their previous work raises concerns. Consequently, it is difficult for the reviewer to endorse the current publication format, particularly due to inconsistencies between the descriptions and figures, which reflect the authors' lack of commitment and possibly a lack of confidence in this extended research.

Here are some concerns and hope the authors can address them:

1. Section 3 should be results and discussion, while Section 4 should present the conclusions. Therefore, the authors should incorporate a more in-depth discussion within Section 3 and ensure a coherent expression of their findings.

2. In 3.1.2, there is a significant difference in particle size observed through DLS. The authors are requested to calculate the corresponding p-values accurately and provide conclusive statements based on the data. Additionally, the authors have not discussed the polydispersity index (PDI) as presented in the results. For instance, the clear increase in PDI after 12 months needs to be addressed in the discussion.

3. Figure a in the SEM images appears blurry. It is suggested that the authors replace this image since, theoretically, low-magnification images should exhibit higher clarity than image b.

4. In section 3.3, the sample at 6 months demonstrated almost complete drug release at the first time point of 4 minutes, which aligns with the authors' statement about the rapid release. However, there is a significant difference in the release trend within the first 30 minutes between the 6-month sample and the 12-month/1-day samples. The authors are expected to discuss this discrepancy.

5. Based on careful consideration of the results, the authors are advised to rethink and revise some words of the conclusion section.

Minor editing of English language required

Author Response

Reviewer 2

Thank you very much for your opinion. Below are listed all of the modifications made in the paper, according to the suggestions, and you can find them with purple color (or orange, if the Reviewer 1 mentioned the section) in the text.

Comments and Suggestions for Authors

The study represents an extension of the authors' previously published work on DPI formulation with MX, aiming to investigate the physico-chemical stability and certain aerodynamic parameters of the formulation over a period of 3-6 months. Although the study is interesting, the authors' lack of meticulousness in completing the manuscript compared to their previous work raises concerns. Consequently, it is difficult for the reviewer to endorse the current publication format, particularly due to inconsistencies between the descriptions and figures, which reflect the authors' lack of commitment and possibly a lack of confidence in this extended research.

Here are some concerns and hope the authors can address them: The following points should be taken into consideration:

  1. Section 3 should be results and discussion, while Section 4 should present the conclusions. Therefore, the authors should incorporate a more in-depth discussion within Section 3 and ensure a coherent expression of their findings.

Thank you for the comment. The result and discussion section were merged and the conclusion section was added.

  1. In 3.1.2, there is a significant difference in particle size observed through DLS. The authors are requested to calculate the corresponding p-values accurately and provide conclusive statements based on the data. Additionally, the authors have not discussed the polydispersity index (PDI) as presented in the results. For instance, the clear increase in PDI after 12 months needs to be addressed in the discussion.

Thank you for the comment. The discussion of the PDI result was modified.

As DLS measured the liquid suspension of the particles. The particles were deagglom-erated and resulted in decreased particle sizes compared to the laser diffraction. The diameters of the dispersed nano spray-dried particles were between 500-700 nm. It predicts that as the dry particles deposit into the lung fluid, and disintegrate into nanoparticles, resulting in larger dissolution and cellular absorption [52]. Although there was an increase in particle diameter, the dry particle size is far more important in terms of lung deposition. Inhomogeneous distribution is shown by the relatively high PDI results (PDI>0.3). However, it is not considered a problem if it does not negatively affect the drug release [39].

  1. Figure a in the SEM images appears blurry. It is suggested that the authors replace this image since, theoretically, low-magnification images should exhibit higher clarity than image b.

Thank you for the comment. The image was changed.

  1. In section 3.3, the sample at 6 months demonstrated almost complete drug release at the first time point of 4 minutes, which aligns with the authors' statement about the rapid release. However, there is a significant difference in the release trend within the first 30 minutes between the 6-month sample and the 12-month/1-day samples. The authors are expected to discuss this discrepancy.

Thank you for the comment. The discussion of the in vitro drug release study was modified.

Figure 5. shows, that the MX released quickly and in larger quantities in the case of the DPI samples than the PM. It is beneficial, because the dissolution rate is the rate‐limiting step for absorption of MX. The nanosized form, the increased surface area, and the partial amorphization of MX promoted a faster dissolution rate [61]. Furthermore, more API was released from the spray-dried samples than the PM because PVA pre-vented particle aggregation and LEU reduced cohesion between the particles. The rap-id release was maintained. Furthermore, the drug release from stored samples was faster, possibly the water absorption on the surface helped the wettability of the parti-cles. The outcomes are beneficial in local treatment, because the mucociliary clearance has minimal time to eliminate the particles [14,62].

  1. Based on careful consideration of the results, the authors are advised to rethink and revise some words of the conclusion section.

Thank you for the comment. The conclusion section was added and revised.

In this study, the stability test of a carrier-free, novel DPI sample containing nanosized non-steroidal anti-inflammatory drug was examined at normal room conditions. After the storage, the formulation presented advantageous characteristics, thanks to the technological steps and the compositions. Wet media milling is one of the most popular method in the pharmaceutical field to produce stable nanosuspension of poorly water-soluble APIs. To optimize time and cost considerations and accurately predict milling performance at higher scales, a variety of modeling techniques could be used in the industry. The spray-drying technique has also been successfully applied at both laboratory and industrial scales. The advantages of a spray-dried powder include easy use and long shelf life stability. The development of novel delivery methods can be a strategy for repositioning medications, therefore it saves money and time for the pharmaceutical industry. The pulmonary route of MX could be an intriguing solution for treating different lung inflammation, which can be caused by SARS-CoV-2 infection, CF and COPD, and NSCLC. The excipients were pulmonary approved materials. During the testing period, the particle size of the remained unchanged, while the particle size of MX in the formulation increased, but did not differ significantly. Furthermore, the partially amorphous property of MX persisted throughout the stability examination. The outcomes of the dissolution test demonstrated, that the initially large amount of drug was released from the samples in the examination time. The results suggest that PVA might inhibit the particles from aggregation and crystallizing. The aerosol performance of the formulated DPIs did not deteriorate. The sample has beneficial FPF and EF results after 12 months. The addition of LEU enhanced the aerosolization of the products. The outcome of this study demonstrates, that the “nano-in.micro” DPI can maintain its quality for an extended period. According to ICH guidelines, further stability investigations are required, such as a test in the final package.

Round 2

Reviewer 1 Report

I suggested the authors to use NGI in next work, and be cautious when the PDI was high. I have no further questions for the current work.